# The Use of a Disposable Umbilical Clamp to Secure an Umbilical Venous Catheter in Neonatal Emergencies—An Experimental Feasibility Study

**DOI:** 10.3390/children8121093

**Published:** 2021-11-26

**Authors:** Bernhard Schwaberger, Christoph Schlatzer, Daniel Freidorfer, Marlies Bruckner, Christina H. Wolfsberger, Lukas P. Mileder, Gerhard Pichler, Berndt Urlesberger

**Affiliations:** 1Division of Neonatology, Department of Pediatrics and Adolescent Medicine, Medical University of Graz, 8036 Graz, Austria; christoph.schlatzer@stud.medunigraz.at (C.S.); marlies.bruckner@medunigraz.at (M.B.); christina.wolfsberger@medunigraz.at (C.H.W.); lukas.mileder@medunigraz.at (L.P.M.); gerhard.pichler@medunigraz.at (G.P.); berndt.urlesberger@medunigraz.at (B.U.); 2Medizinercorps Graz, Austrian Red Cross Federal Association Styria, 8010 Graz, Austria; Daniel.Freidorfer@st.roteskreuz.at

**Keywords:** (secure method for) umbilical venous catheter (UVC), UVC securement technique, neonatal resuscitation, neonatal emergency, disposable umbilical clamp, vascular access, newborn

## Abstract

Recent guidelines recommend the umbilical venous catheter (UVC) as the optimal vascular access method during neonatal resuscitation. In emergencies the UVC securement may be challenging and time-consuming. This experimental study was designed to test the feasibility of new concepts for the UVC securement. Umbilical cord remnants were catheterized with peripheral catheters and secured with disposable umbilical clamps. Three different securement techniques were investigated. Secure 1: the disposable umbilical clamp was closed at the level of the inserted catheter. Secure 2: the clamp was closed at the junction of the catheter and plastic wings. Secure 3: the setting of Secure 2 was combined with an umbilical tape. The main outcomes were the feasibility of fluid administration and the maximum force to release the securement. This study shows that inserting peripheral catheters into the umbilical vein and securing them with disposable umbilical clamps is feasible. Rates of lumen obstruction and the effectiveness of the securement were superior with Secure 2 and 3 compared to Secure 1. This new approach may be a rewarding option for umbilical venous catheterization and securement particularly in low-resource settings and for staff with limited experience in neonatal emergencies. However, although promising, these results need to be confirmed in clinical trials before being introduced into clinical practice.

## 1. Introduction

The umbilical venous catheter (UVC) is considered “the most quickly accessible direct intravenous route” into the newborn [1,2]. Thus, recent guidelines recommend the UVC as the optimal vascular access method for drug administration during neonatal resuscitations [3,4,5]. Despite its frequent use, there is still a lack of knowledge on the best technique for this catheterization and UVC securement in emergency situations. The proper position of a centrally positioned UVC should be confirmed sonographically or radiographically, although this might be challenging during an actual resuscitation [6,7,8]. In the case of UVC malpositioning, there is the risk of adverse events including infusing drugs directly into the liver veins, potentially resulting in hepatic injuries [9,10,11,12], and furthermore, cardiac complications such as arrhythmias or cardiac tamponades [13,14]. Therefore, in neonatal emergencies it is recommended to insert the UVC only two to five cm below the skin (and even less for premature infants) until the blood can be aspirated gently via a syringe [1,2,15]. For the UVC securement, in the 7th edition of the *Textbook of Neonatal Resuscitation* a combination of suturing and taping of the UVC, or, alternatively, the use of a clear adhesive dressing is recommended. Nevertheless, both techniques require some time and may not be easily realized during emergencies [1]. However, due to a considerable risk of the accidental dislocation of the UVC during resuscitation, there is the need for an effective securement method.

In July 2018, a physician-staffed Emergency Medical Service (EMS) was faced with an unplanned out-of-hospital delivery of an extremely low-birth-weight infant of 27 weeks’ gestation weighing approximately 900 g in the urban area of Graz, Austria [16]. During the neonatal resuscitation, epinephrine and a fluid administration was required, and an umbilical venous catheterization using a 22-gauge peripheral catheter was successfully performed. For the securement of the UVC, the EMS staff spontaneously used a disposable umbilical clamp. Epinephrine and a fluid bolus administration was feasible, and the securement was deemed very effective.

To investigate this concept of umbilical venous catheterization using a standard peripheral catheter and securement with a disposable umbilical clamp, we decided to perform this experimental feasibility study. The feasibility of the fluid administration and the force needed to release the securement was measured to detect relevant obstructions of the catheter lumen caused by the securement technique and to evaluate the effectiveness of the securement. The aim was to find a feasible and effective technique for neonatal emergencies, which could be performed even in low-resource settings (e.g., the out-of-hospital setting) using standard equipment.

## 2. Materials and Methods

This experimental feasibility study was conducted at the Division of Neonatology, Department of Paediatrics and Adolescent Medicine, Medical University of Graz, from July to August 2019. Human umbilical cords, which were already separated from the newborn infants, were used for this study. We included umbilical cord remnants from both premature and full-term infants, without a predefined number of umbilical cord remnants from premature and full-term infants.

Immediately after their separation from the newborn infant, the umbilical cord remnants were perpendicularly cut with a scalpel. The cut surface was cleaned using saline solution to identify the umbilical vein. Any visible clots at the meatus of the vein were gently removed. The umbilical vein was then catheterized with a standard peripheral catheter (B. Braun, Melsungen, Germany), using an 18-gauge catheter for full-term infants and a 20-gauge catheter for premature infants with <37 + 0 weeks’ gestation. Whenever the catheterization with an 18-gauge catheter was not feasible, another attempt was made using a 20-gauge catheter. The catheter was inserted into the umbilical vein as far as possible until the plastic wings of the catheter adjoined the cut surface of the umbilical cord.

For the securement of the inserted catheter a disposable umbilical clamp (pfm medical, Cologne, Germany) was used. Three different securement techniques were investigated and compared: Secure 1, Secure 2 and Secure 3. We randomly assigned the umbilical cord remnants to one of the securement techniques and aimed for 20 successful catheterizations with each technique. For the random assignment we did not stratify for premature and full-term infants.

**Secure 1:** The disposable umbilical clamp was closed at the level of the inserted transparent catheter (Figure 1A and Figure 2A).

**Secure 2:** The disposable umbilical clamp was closed at the junction of the transparent catheter and the colored plastic wings (Figure 1A,B and Figure 2B).

**Secure 3:** The disposable umbilical clamp was used identically to that in Secure 2, but additionally an “umbilical tape” (Medi-Loop Sterile Surgical Vessel Loops, Medline Industries, Warrington, United Kingdom) was placed around the umbilical cord at the level of the transparent catheter (Figure 1A,B and Figure 2C).

The main outcomes of this study were (i) the feasibility of the fluid administration and (ii) the effectiveness of the three UVC securement techniques. 

To test the feasibility of the fluid administration, a predefined bolus of 10 mL 0.9% saline solution per kg of the body weight of the corresponding newborn infant was continuously administered by hand via the inserted catheter using disposable syringes (Chirana T. Injecta, Stará Turá, Slovakia). We aimed at infusing the entire fluid bolus within one minute. The free end of the umbilical cord remnant was positioned in a measuring cup, and the infused fluid was thereby collected (Figure 3A). The other end of the umbilical cord remnant with the inserted catheter and the connected syringe was held outside of the measuring cup beneath the level of its opening to prevent retrogradely leaking fluid to drip into the measuring cup. To record the fluid level in the measuring cup, the umbilical cord remnant was removed after the one-minute administration and held in position to allow the fluid to drip off into the cup for another 30 s (Figure 3B). The ratio of the within-one-minute actually administered fluid volume to the predefined volume was calculated afterward to evaluate the feasibility of the fluid administration. There were two factors that might have affected the feasibility of the fluid administration: obstruction and leakage. Failing to purge any fluid from the syringes was defined as a complete obstruction of the catheter lumen due to the securement technique. A fluid amount (that was not equal to the entire predefined volume) that remained in the syringes after the one-minute fluid administration indicated a partial obstruction. Leakage was defined by the fluid amount from the predefined bolus that was not collected in the measuring cup and that did not remain in the syringes after the one-minute fluid administration.

To measure the effectiveness of the three UVC securement techniques, an electronic spring scale (Dr. Meter, United Kingdom) was connected to a prepared disposable syringe and to the catheter via a Luer lock connection. By slowly pulling the disposable umbilical clamp, the force to release the securement was measured (Figure 4). To determine the maximum force value on the spring scale’s display, the display was filmed with a digital camera and the maximum force value was identified retrospectively in a slow-motion video analysis.

Data collected included: the actually infused fluid volume; complete obstruction of the catheter lumen; remaining fluid amounts in the syringes after the one-minute fluid administration; leakage; maximum force required to release the securement; size of the peripheral catheter (20 or 18 gauge); and demographic data, including the gestational age and birth weight of the corresponding newborn infant. The parameters are presented as mean ± standard deviation (SD), median and interquartile range (IQR) or count (proportion), as appropriate. For the gestational age and birth weight the range is provided additionally to highlight the broad spectrum of newborn infants whose umbilical cord remnants were used. Data analysis was conducted with SPSS 26.0.0.1 (IBM, Armonk, NY, USA). Comparisons between the securement techniques were made using the chi-square test, Student’s *t*-test or Mann–Whitney U-test, as appropriate. A *p*-value < 0.05 was considered statistically significant.

## 3. Results

A total of 65 umbilical cord remnants were prepared for umbilical venous catheterization. Five had to be excluded: in four of them the UVC could not be inserted far enough into the umbilical vein and thus a securement was not feasible. Another one was excluded due to the extravasation of the infused fluid bolus into the Wharton jelly. Thus, data on 20 umbilical cord remnants per securement technique were finally analyzed.

Umbilical cord remnants from 40 (67%) full-term infants and 20 (33%) premature infants with <37 + 0 weeks’ gestation were included. The mean (SD) birth weight of the corresponding newborn infants was 2.86 (0.85) kg (range 0.35–4.42), and the median gestational age was 36.9 (IQR 33.9–39.9) weeks (range 26.1–40.6). 

### 3.1. Size of Catheter

In only 58% (23 of 40 cases) of the umbilical cord remnants from full-term infants it was feasible to insert an 18-gauge catheter into the umbilical vein. There was no significant difference in the ratio of the actually administered fluid to the predefined volume depending on the size of the peripheral catheter: 100% (IQR 83–100) with the 18-gauge catheter and 93% (IQR 71–100) with the 20-gauge catheter (*p* = 0.64).

### 3.2. Feasibility of Fluid Administration 

A complete obstruction of the UVC lumen was observed six times (30%) with Secure 1, never (0%) with Secure 2 and once (5%) with Secure 3 (Secure 1 vs. 2, *p* < 0.01; Secure 1 vs. 3, *p* = 0.04; Secure 2 vs. 3, *p* = 0.31). A partial obstruction was observed twice (10%) with Secure 1, never (0%) with Secure 2 and twice (10%) with Secure 3 (Secure 1 vs. 2, *p* = 0.15; Secure 1 vs. 3, *p* = 1.00; Secure 2 vs. 3, *p* = 0.15). 

The ratio of the within-one-minute actually administered fluid volume to the predefined volume was 97% (IQR 0–100%) with Secure 1, compared to 90% (IQR 69–100%) with Secure 2 and 95% (IQR 89–100%) with Secure 3. There were no significant differences between these three securement techniques (Secure 1 vs. 2, *p* = 0.27; Secure 1 vs. 3, *p* = 0.21; Secure 2 vs. 3, *p* = 0.71).

The leakage was calculated to be 0 (IQR 0–0) mL with Secure 1, compared to 1.5 (IQR 0–3.0) mL with Secure 2 and 3.0 (IQR 0–7.5) mL with Secure 3 (Secure 1 vs. 2, *p* = 0.05; Secure 1 vs. 3, *p* < 0.01; Secure 2 vs. 3, *p* = 0.27).

### 3.3. Effectiveness of the Securement

The maximal force required to release the securement was 4.6 N (IQR 3.9–6.0) with Secure 1, 50.1 N (IQR 38.9–70.6) with Secure 2 and 65.9 N (IQR 56.5–68.9) with Secure 3 (Secure 1 vs.2, *p* < 0.01; Secure 1 vs.3, *p* < 0.01; Secure 2 vs. 3, *p* = 0.22).

## 4. Discussion

This experimental feasibility study was designed to test a new method for gaining vascular access in neonatal emergencies by inserting a standard peripheral catheter into the umbilical vein and securing it with a disposable umbilical clamp. The study demonstrates that using a disposable umbilical clamp for a UVC securement is feasible and effective. In our experience, this approach is simple and can be performed quickly. Thus, it may be a rewarding option, particularly for staff with limited experience in neonatal resuscitation.

In this study three different UVC securement techniques using disposable umbilical clamps were compared. The feasibility of the fluid administration was not different between the three techniques. However, the fluid administration was impeded by both obstructions and leakages, and concerning these factors, we observed relevant differences between the three securement techniques.

A catheter obstruction may be caused by closing the disposable umbilical clamp and thereby compressing the catheter lumen. A complete lumen obstruction can be distinguished from a partial lumen obstruction as defined in the methods section. The rate of complete catheter obstruction was significantly higher with Secure 1 compared to Secure 2 and Secure 3, which is clinically most relevant, since in cases of a complete catheter obstruction neither epinephrine nor fluids could be administered successfully during neonatal resuscitation. For Secure 1, the disposable umbilical clamp was closed at the level of the inserted transparent catheter. The transparent part of the catheter is obviously more flexible and, thus, compressible compared to the junction of the transparent catheter and the colored plastic wings, which is the position of the closed umbilical clamp in Secure 2 and Secure 3. Furthermore, we observed cases of partial obstruction not only with Secure 1 but also with Secure 3, which may impede the quick application of a fluid bolus. However, despite a partial obstruction, administering epinephrine, including a fluid flush of 1–2 mL, is still feasible within seconds, and a fluid bolus administration may also be possible even though slower infusion rates must be accepted. Based on these findings, the use of both Secure 2 and Secure 3 seem to be reasonable for UVC securements. 

The leakage was significantly lower with Secure 1 compared to Secure 2 and Secure 3, which explains why the overall feasibility of the fluid administration was not different between the three techniques, despite higher obstruction rates with Secure 1. Leakages mainly occurred during the first seconds of the one-minute fluid administration with high purging pressures at the beginning. As soon as the umbilical vein was free from obstructions over its entire length and the purging pressure could be reduced, there was no retrogradely leaking fluid in most cases. Therefore, we speculate that the measured leakage was artificially high with Secure 2 and Secure 3 and probably caused by the experimental set-up of the study. Furthermore, the leakage was per definition zero in cases of complete obstruction, and due to the high complete obstruction rate the median leakage was probably underestimated with Secure 1. Hence, in our opinion the observed median leakage may not provide sufficient information to assess the effectiveness of the bleeding control. To answer this question, clinical studies are certainly needed. 

The effectiveness of the UVC securement (measured by the maximal force required to release the securement) was significantly higher with Secure 2 and Secure 3 compared to Secure 1. In our experience, the maximal forces needed were rather high, especially with Secure 2 and Secure 3, compared to other previously described securement techniques [17]. However, there are no data available that would allow a direct numerical comparison with our data. Indeed, the effectiveness of the securement technique may be particularly relevant during neonatal transport and in low-resource settings, in which the patients frequently need to be transferred and/or repositioned. Therefore, Secure 2 or Secure 3 should be considered for UVC securement especially in such circumstances.

Different techniques for UVC securement have been described before, which use tapes, other adhesive materials or sutures [17]. However, immediately after birth the newborn’s skin may be wet and covered with vernix, and tapes and adhesive materials may not adhere properly. Using suture needles in such situations is accompanied by a related risk of needlestick injuries, and may be difficult to perform in particular if chest compressions are required or during out-of-hospital situations. In addition, traditional techniques for UVC securement are technically challenging and relatively time-consuming during neonatal emergencies. A simulation-based study has shown that UVC placements and securements during neonatal resuscitations take approximately six minutes, and thus may severely delay the intravenous administration of epinephrine [18]. However, it is recommended that one person should hold the successfully inserted UVC in place, while another person administers the first dose of epinephrine and/or a fluid bolus during resuscitation [1]. The securement of the UVC for continued vascular access should be performed only after the first emergency drugs have been successfully administered [1]. Based on our experience, the newly introduced securement techniques can be performed quickly and with ease, although we did not measure time intervals, since our study was not a simulation-based study but an experimental feasibility study. Nonetheless, this new approach may be a rewarding option for UVC securement particularly during neonatal resuscitations.

Recent guidelines [3,4,5] recommend the UVC for drug administration during neonatal resuscitations, which is rarely performed (required in only 0.12% of all deliveries), requires significant skill and may be further impeded by space constraints for the resuscitation team [19]. Alternatively, vascular access may be achieved via a peripheral vein [20] or intraosseously [21,22]. Outside of the delivery room setting, the intraosseous access is being used more frequently by health care providers with limited experience and training in neonatal resuscitation, but with experience with intraosseous needle placement (i.e., EMS staff) [3]. With the herein presented new approach to umbilical venous catheterization and securement, which can be performed easily and quickly with standard equipment, the UVC might gain significance also in the abovementioned settings. Non-neonatologist health care providers might benefit from the introduction of the new technique and the potentially increased utilization of the UVC in the future, since adverse effect rates attributable to emergency umbilical venous catheterization might be lower compared to the intraosseous access. Further, a UVC can be achieved even in extremely low-birth-weight infants, while most of the available devices for intraosseous access have a higher minimum weight limit [21]. Indeed, personnel should be trained in umbilical venous catheterization periodically, even with the simple new approach for securement, ideally with real umbilical cords due to the higher physical and functional fidelity [23,24]. 

### Limitations and Disadvantages

The main limitation of this study is its experimental character, which implies that some clinical research questions (e.g., the effectiveness of bleeding control) cannot be resolved. Although the feasibility of the new securement techniques was demonstrated experimentally, clinical studies are required to confirm our results, before this approach can be introduced safely into clinical practice. Furthermore, future research should include red blood cell transfusions, since in our study 0.9% saline solution was administered through the UVC, and different viscosities may have an impact on the feasibility of the new approach.

One disadvantage of the newly introduced securement techniques is the risk of catheter obstructions caused by the disposable umbilical clamp. Once the disposable umbilical clamp is closed, it may irreversibly compress the lumen of the peripheral plastic catheter. Using a metal cannula (e.g., a bulb-headed probe) instead of the flexible peripheral catheter could help preventing such lumen obstructions. Alternatively, reusable plastic clamps that spring open again when released could be used. However, we aimed to test the concept of using peripheral catheters in combination with disposable umbilical clamps for UVC securement, since these devices are generally available even in low-resource settings and belong to the standard equipment (e.g., in ambulance vehicles).

Considering a recent animal study, in which a higher flush volume after the first dose of epinephrine was shown to be beneficial during neonatal resuscitation [25], there is likely need for an even higher volume of saline flush following epinephrine with the new UVC techniques compared to the centrally placed UVC, because of the additional length of the umbilical vein to be flushed.

Another disadvantage is that the integrity of the umbilical arteries will likely be compromised following the placement of the umbilical clamp, and the umbilical arterial access and placement of a long-term umbilical venous catheter would need to be performed distal to the clamp placement. Therefore, there should be enough umbilical cord remaining between the clamp and the umbilicus to ensure that future access to the umbilical vessels will be possible.

## 5. Conclusions

Inserting a standard peripheral catheter (18 or 20 gauge) into the umbilical vein and securing it by using a disposable umbilical clamp was feasible with all three investigated securement techniques (Secure 1–3). Rates of complete catheter lumen obstruction and the effectiveness of securement was superior with Secure 2 and 3 compared to Secure 1. Still, these results need to be confirmed in clinical trials before being introduced into clinical routines. During neonatal resuscitations, the new approach may be a rewarding option for umbilical venous catheterizations and UVC securements, particularly in low-resource settings and for staff with limited experience in neonatal emergencies. 

## Figures and Tables

**Figure 1 children-08-01093-f001:**
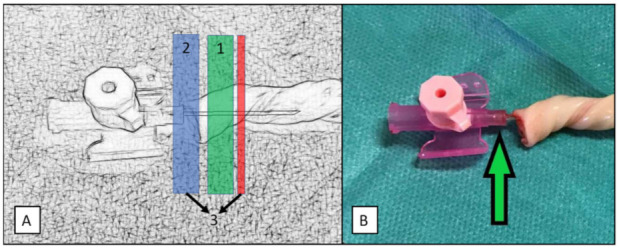
(**A**): Graphic illustration of the three different securement techniques (Secure 1–3): a human umbilical cord remnant was catheterized with a peripheral catheter. For Secure 1, a disposable umbilical clamp was closed in the area of the green box (1) at the level of the inserted transparent part of the catheter. For Secure 2 and Secure 3, a disposable umbilical clamp was closed in the area of the blue box (2) at the junction of the transparent catheter and the plastic wings. For Secure 3, an umbilical tape was additionally placed around the umbilical cord in the area of the red box at the level of the transparent catheter (3). (**B**): The green arrow indicates the junction of the transparent catheter and the colored plastic wings of a 20-gauge peripheral catheter. The disposable umbilical clamp was closed at the level of this junction in Secure 2 and Secure 3.

**Figure 2 children-08-01093-f002:**
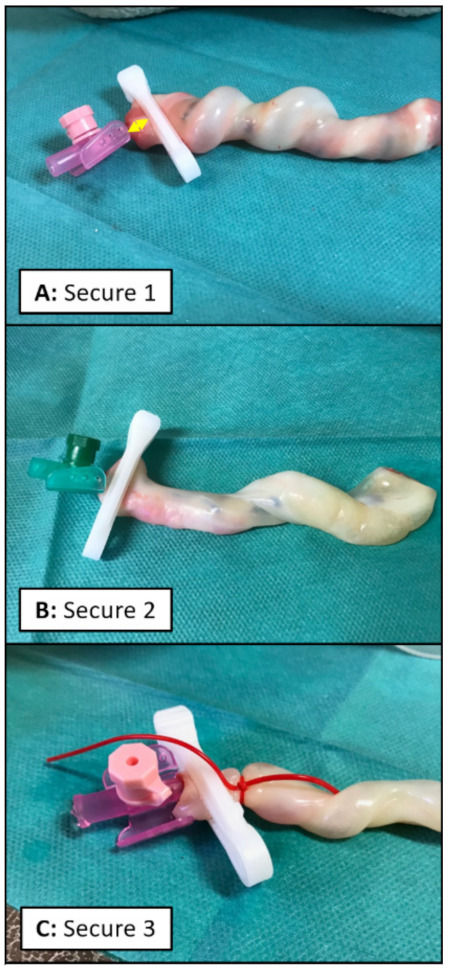
(**A**): Secure 1: a 20-gauge peripheral catheter inserted into the umbilical vein secured by a disposable umbilical clamp closed at the level of the transparent catheter. The yellow arrows indicate the distance between the colored plastic wings and the disposable umbilical clamp, which is longer in Secure 1 compared to Secure 2 and Secure 3. (**B**): Secure 2: an 18-gauge peripheral catheter inserted into the umbilical vein secured by a disposable umbilical clamp closed at the junction of the transparent catheter and the colored plastic wings. (**C**): Secure 3: a 20-gauge peripheral catheter inserted into the umbilical vein secured by a disposable umbilical clamp closed at the junction of the transparent catheter and the colored plastic wings, and by an additional umbilical tape placed around the umbilical cord at the level of the transparent catheter.

**Figure 3 children-08-01093-f003:**
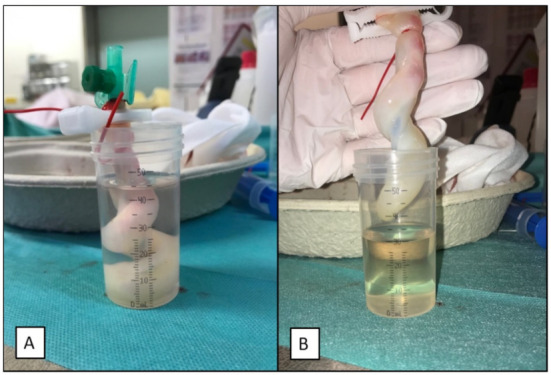
(**A**): The predefined fluid bolus of 0.9% saline solution was administered over one minute via the inserted and secured catheter. The free end of the umbilical cord remnant was positioned into a measuring cup, and the infused fluid was thereby collected. (**B**): To record the fluid level in the measuring cup, the umbilical cord remnant was removed and held in position to allow the fluid to drip off into the cup for another 30 s.

**Figure 4 children-08-01093-f004:**
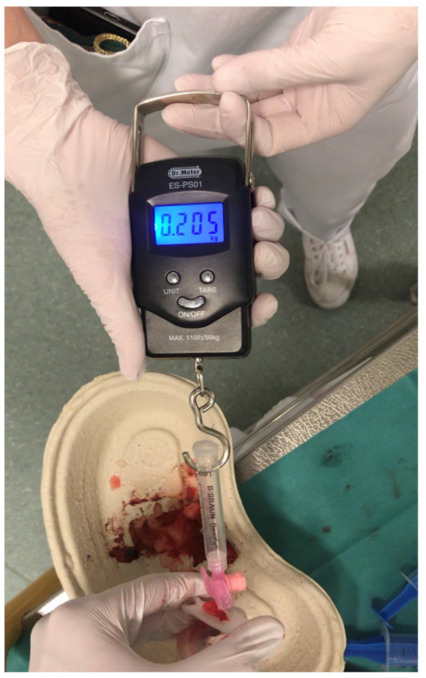
An electronic spring scale was connected to a prepared disposable syringe and to the catheter via a Luer lock connection. By slowly pulling the disposable umbilical clamp, the force to release the securement was measured.

## Data Availability

The data presented in this study are available in this article.

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
