# Peer review of "The Use of a Disposable Umbilical Clamp to Secure an Umbilical Venous Catheter in Neonatal Emergencies—An Experimental Feasibility Study"

_children, 2021, doi:10.3390/children8121093_

Round 1

Reviewer 1 Report

In this manuscript, the authors describe a novel approach to secure peripheral venous catheters onto the umbilical vein in emergent situations in resource-limited settings by using a disposal umbilical clamp. Using human cut umbilical cords, three different securement methods were evaluated: (1) using the umbilical clamp along the catheter, (2) using the clamp at the base of the wings and 3) tying an umbilical tape distal to the clamp as secured in (2). 

The authors assess patency of the lumen by injecting a 10 ml/kg NS bolus within one minute. 

They observed when the clamp is closed over the lumen of the catheter that 30% of cases had complete obstruction, but that with methods 2 and 3, patency was maintained in 100%. The figures are very helpful. 

The idea is novel and this method could be helpful in resource-limited settings where a conventional umbilical catheter cannot be inserted owing to lack of expertise or specialized equipment. 

Comments/suggestions
Indwelling venous catheter should be changed to peripheral catheter, which is more descriptive. 

The integrity of the umbilical arteries will likely be compromised following placement of the umbilical clamp and umbilical arterial access and placement of a long-term umbilical venous catheter would need to be done distal to the clamp placement. The authors should consider discussing that there should be enough umbilical cord remaining between the clamp and the umbilicus to ensure that future access of the umbilical vessels will be possible.   

Author Response

In this manuscript, the authors describe a novel approach to secure peripheral venous catheters onto the umbilical vein in emergent situations in resource-limited settings by using a disposal umbilical clamp. Using human cut umbilical cords, three different securement methods were evaluated: (1) using the umbilical clamp along the catheter, (2) using the clamp at the base of the wings and 3) tying an umbilical tape distal to the clamp as secured in (2). 

The authors assess patency of the lumen by injecting a 10 ml/kg NS bolus within one minute. 

They observed when the clamp is closed over the lumen of the catheter that 30% of cases had complete obstruction, but that with methods 2 and 3, patency was maintained in 100%. The figures are very helpful. 

The idea is novel and this method could be helpful in resource-limited settings where a conventional umbilical catheter cannot be inserted owing to lack of expertise or specialized equipment. 

Comments/suggestions
Indwelling venous catheter should be changed to peripheral catheter, which is more descriptive. 

WE CHANGED THE WORDING TO "PERIPHERAL CATHETER" THROUGOUT THE ENTIRE MANUSCRIPT ACCORDING TO THE REVIEWER'S SUGGSTION.

The integrity of the umbilical arteries will likely be compromised following placement of the umbilical clamp and umbilical arterial access and placement of a long-term umbilical venous catheter would need to be done distal to the clamp placement. The authors should consider discussing that there should be enough umbilical cord remaining between the clamp and the umbilicus to ensure that future access of the umbilical vessels will be possible. 

WE INCLUDED THE FOLLOWING SECTENCES WITHIN THE DISCUSSION SECTION:

"Another disadvantage is that the integrity of the umbilical arteries will likely be compromised following placement of the umbilical clamp, and umbilical arterial access and placement of a long-term umbilical venous catheter would need to be done distal to the clamp placement. Therefore, there should be enough umbilical cord remaining between the clamp and the umbilicus to ensure that future access of the umbilical vessels will be possible."

THANK YOU FOR THE VALUABLE COMMENTS AND SUGGSTIONS WHICH HELPED TO IMPROVE THE MANUSCRIPT!

Reviewer 2 Report

Overall, a very interesting feasibilty study that clearly provides justification for future clinical research to rigourously evaluate this basic, yet novel securement method. This research will have important practice implications for low-resource centres and out-of-hospital neonatal resuscitations.

The manuscript is well written and the images of the experimental set-up add value.

Some minor recommendations/considerations to improve the manuscript are as follows:

  1. In the background section mention the risk of accidental dislodgment of the UVC during resuscitation, hence the need for an effective securement method.
  2. Blood (high viscosity) is often adminsitered during neonatal resuscitation and was not tested, therefore this should be a limitation of the study and a recommendation for future research.
  3. How did the researchers ensure the umbilical tape was tightened consitently amongst the cords for Secure 3? Could there be variation on how tight they were? Can this be measured? Some institutions may also use suture material to tie.

Formating: line 241 "In our experience", line 275 long sentence

I look forward to reading the results of the clinical studies.

Author Response

Overall, a very interesting feasibilty study that clearly provides justification for future clinical research to rigourously evaluate this basic, yet novel securement method. This research will have important practice implications for low-resource centres and out-of-hospital neonatal resuscitations.

The manuscript is well written and the images of the experimental set-up add value.

Some minor recommendations/considerations to improve the manuscript are as follows:

  1. In the background section mention the risk of accidental dislodgment of the UVC during resuscitation, hence the need for an effective securement method.

WE INCLUDED THE FOLLOWING SENTENCE TO THE INTRODUCTION:
"However, due to a considerable risk of accidental dislocation of the UVC during resuscitation, there is the need for an effective securement method."

  1. Blood (high viscosity) is often adminsitered during neonatal resuscitation and was not tested, therefore this should be a limitation of the study and a recommendation for future research.

WE INCLUDED THE FOLLOWING SENTENCE TO THE LIMITATION SECTION:
“Furthermore, future research should include red blood cell transfusions, since in our study 0.9% saline solution was administered through the UVC, and different viscosities may have an impact on the feasibility of the new approach.”

  1. How did the researchers ensure the umbilical tape was tightened consitently amongst the cords for Secure 3? Could there be variation on how tight they were? Can this be measured? Some institutions may also use suture material to tie.

THANK YOU FOR THIS QUESTION. THE UMBILICAL TAPE WAS ALWAYS TIGHTEND BY A SINGLE INVESTIGATOR (TO EXCLUDE ANY INTER-INVESTIGATOR VARIABILTY). WE THINK THAT THERE IS NO POSSIBILITY TO MEASURE THIS ISSUE. INDEED, SUTURE MATERIAL COULD ALSO BE USED INSTEAD, BUT WE DECIDED TO GO WITH UMBILICAL TAPES AS IT IS THE WAY WE DO IT IN OUR CENTER.

Formating: line 241 "In our experience", line 275 long sentence

WE CHANGED THE WORDING TO "IN OUR EXPERIENCE" (LINE 241)

LINE 275: TO MAKE THE LONG SENTENCE MORE READIBLE, WE SPLIT THE SENTENCE:
"Non-neonatologist health care providers might benefit from the introduction of the new technique and the potentially increased utilization of UVC in the future, since adverse effect rates attributable to emergency umbilical venous catheterization might be lower compared to intraosseous access. Further, an UVC can be achieved even in extremely low birth weight infants, while most of the available devices for intraosseous access have a higher minimum weight limit [21]"

I look forward to reading the results of the clinical studies.

THANK YOU FOR THE VALUABLE COMMENTS AND SUGGSTIONS WHICH HELPED TO IMPROVE THE MANUSCRIPT!